# Impact of Inflammation on Voriconazole Exposure in Critically ill Patients Affected by Probable COVID-19-Associated Pulmonary Aspergillosis

**DOI:** 10.3390/antibiotics12040764

**Published:** 2023-04-16

**Authors:** Milo Gatti, Giacomo Fornaro, Zeno Pasquini, Andrea Zanoni, Michele Bartoletti, Pierluigi Viale, Federico Pea

**Affiliations:** 1Department of Medical and Surgical Sciences, Alma Mater Studiorum University of Bologna, 40138 Bologna, Italy; milo.gatti2@unibo.it (M.G.); pierluigi.viale@unibo.it (P.V.); 2Clinical Pharmacology Unit, Department for Integrated Infectious Risk Management, IRCCS Azienda Ospedaliero-Universitaria di Bologna, 40138 Bologna, Italy; 3Infectious Diseases Unit, Department for Integrated Infectious Risk Management, IRCCS Azienda Ospedaliero-Universitaria di Bologna, 40138 Bologna, Italy; giacomofornaro1990@gmail.com (G.F.); zeno.pasquini@gmail.com (Z.P.); 4Division of Anesthesiology, Department of Anesthesia and Intensive Care, IRCCS Azienda Ospedaliero-Universitaria di Bologna, 40138 Bologna, Italy; andrea.zanoni@aosp.bo.it; 5IRCCS Humanitas Research Hospital, 20089 Milan, Italy; michele.bartoletti@hunimed.eu; 6Department of Biomedical Sciences, Humanitas University, 20089 Milan, Italy

**Keywords:** voriconazole, COVID-associated pulmonary aspergillosis, inflammation, C-reactive protein, cytochrome P450 downregulation, therapeutic drug monitoring

## Abstract

(1) Background: To explore the impact of the degree of inflammation on voriconazole exposure in critically ill patients affected by COVID-associated pulmonary aspergillosis (CAPA); (2) Methods: Critically ill patients receiving TDM-guided voriconazole for the management of proven or probable CAPA between January 2021 and December 2022 were included. The concentration/dose ratio (C/D) was used as a surrogate marker of voriconazole total clearance. A receiving operating characteristic (ROC) curve analysis was performed by using C-reactive protein (CRP) or procalcitonin (PCT) values as the test variable and voriconazole C/D ratio > 0.375 (equivalent to a trough concentration [C_min_] value of 3 mg/L normalized to the maintenance dose of 8 mg/kg/day) as the state variable. Area under the curve (AUC) and 95% confidence interval (CI) were calculated; (3) Results: Overall, 50 patients were included. The median average voriconazole C_min_ was 2.47 (1.75–3.33) mg/L. The median (IQR) voriconazole concentration/dose ratio (C/D) was 0.29 (0.14–0.46). A CRP value > 11.46 mg/dL was associated with the achievement of voriconazole C_min_ > 3 mg/L, with an AUC of 0.667 (95% CI 0.593–0.735; *p* < 0.001). A PCT value > 0.3 ng/mL was associated with the attainment of voriconazole C_min_ > 3 mg/L (AUC 0.651; 95% CI 0.572–0.725; *p* = 0.0015). (4) Conclusions: Our findings suggest that in critically ill patients with CAPA, CRP and PCT values above the identified thresholds may cause the downregulation of voriconazole metabolism and favor voriconazole overexposure, leading to potentially toxic concentrations.

## 1. Introduction

The SARS-CoV-2 pandemic has been responsible for the most intensive care unit (ICU) admission in the last three years, accounting for remarkable morbidity and mortality [1]. Bacterial and fungal superinfections have been widely reported in critically ill COVID-19 patients, with prevalence ranging from 16% to 40% [2,3,4].

COVID-19-associated pulmonary aspergillosis (CAPA) emerged as a severe fungal superinfection in ICU-admitted patients, affecting up to 40% of patients undergoing mechanical ventilation [5,6]. COVID-19 itself, coupled with the use of immunomodulatory agents (i.e., corticosteroids and/or tocilizumab), may strongly compromise the immune system, leading to the emergence of opportunistic superinfections [6]. This clinical picture was similar to those of invasive pulmonary aspergillosis previously observed in critically ill patients affected by severe influenza pneumonia requiring ICU management [7]. Notably, the occurrence of CAPA seems to increase the mortality rate (i.e., greater than 50%) among COVID-19-patients [5,6,8]. In previous studies, the treatment of CAPA with voriconazole was associated with a trend toward a decrease in mortality rate [5,9].

Implementing a therapeutic drug monitoring (TDM) strategy for guiding voriconazole dosing should be considered mandatory among ICU patients due to the well-recognized huge inter- and intra-individual pharmacokinetic variability that may make unpredictable drug exposure at fixed dosing regimens [10,11]. Specifically, a recent international position paper strongly recommended the routine implementation of a TDM-guided strategy when voriconazole is administered in critically ill patients [10]. Indeed, the application of a TDM-guided strategy showed to improve clinical efficacy and safety of voriconazole, by minimizing the risk of treatment withdrawal because of adverse events [12]. Although real-world evidence investigating the role of a TDM-guided strategy for voriconazole in critically ill patients is limited, the occurrence of underexposure associated with negative clinical outcome was reported [13].

Among the different factors possibly impacting on voriconazole exposure, recent evidence found that the degree of inflammation may significantly affect voriconazole concentrations [14,15,16,17]. Voriconazole metabolism is mediated by CYP2C9, CYP2C19, and CYP3A4 [18]. Indeed, an ever-growing number of studies showed that several pro-inflammatory cytokines, especially IL-6, may moderately downregulate the activity of CYP3A4 and weakly-to-moderately downregulate those of CYP2C9 and CYP2C19 [19,20]. Considering that inflammation represents a common feature that shows remarkable proportion among ICU patients, a relevant impact on the occurrence of voriconazole overexposure and toxicity could not be ruled out [14]. However, no studies investigated this issue in the challenging scenario of CAPA.

The aim of this study was to explore the impact of the degree of inflammation by using C-reactive protein (CRP) and procalcitonin (PCT) serum levels as inflammatory biomarkers on voriconazole exposure in critically ill patients affected by CAPA.

## 2. Results

Overall, a total of 50 critically ill patients received TDM-guided voriconazole for treating probable CAPA during the study period (Table 1).

Median (interquartile [IQR]) age was 65.5 (60.0–73.75) years, with a slight male preponderance (54.0%). Thirty-three out of 50 patients (66.0%) had comorbidities, with obesity (30.0%), type 2 diabetes mellitus (20.0%), and cancer (16.0%) being the most frequent ones. Most patients (90.0%) underwent mechanical ventilation, whereas vasopressor support was needed in 44.0% of cases. Extracorporeal membrane oxygenation (ECMO) was applied in three patients. COVID-19 treatment was implemented in all of the included critically ill patients, and the most frequent administered drugs were dexamethasone (88.0%), low molecular weight heparin (52.0%), and tocilizumab (44.0%).

The median (IQR) baseline galactomannan (GM) index on bronchoalveolar lavage (BAL) before starting voriconazole treatment was 3.41 (1.84–4.63). *Aspergillus* spp. was isolated on BAL cultures in nine cases (18.0%), with *Aspergillus fumigatus*, *Aspergillus flavus*, *Aspergillus niger*, and *Aspergillus fumigatus* and *flavus* reported in five, two, and one case each, respectively. Bacterial superinfections were present in 42 out of 50 patients (84.0%), with *Pseudomonas aeruginosa* (24.0%), *Klebsiella pneumoniae* (24.0%), and *Enterococcus faecalis* (16.0%) being the most prevalent clinical isolates. Gram-negative pathogens were carbapenem-resistant in 11 cases (22.0%).

The median (IQR) number of TDM assessments of voriconazole trough concentrations (C_min_) per patient was 3 (2–6). The median (IQR) average voriconazole C_min_ was 2.47 (1.75–3.33) mg/L, whereas the median (IQR) average voriconazole daily dose was 7.69 (6.99–8.13) mg/kg. The median (IQR) voriconazole concentration/dose ratio (C/D) was 0.29 (0.14–0.46). Potentially interacting medications were co-administered in 68.0% of cases (34/50), but none of these was a strong inhibitor or inducer of CYP3A4 or 2C9/2C19 isoenzymes.

A total of 182 paired voriconazole C_min_-serum C-reactive protein (CRP) determinations were performed in the included patients. Receiving operating characteristics (ROC) curve analysis showed that the CRP value > 11.46 mg/dL was associated with voriconazole C_min_ > 3 mg/L, with a sensitivity of 52.2% (95% confidence interval [CI] 39.7–64.6%) and specificity of 85.2% (95% CI 77.4–91.1%). The area under curve (AUC) was 0.667 (95% CI 0.593–0.735; *p* < 0.001; Figure 1).

A total of 161 paired voriconazole C_min_-serum procalcitonin (PCT) determinations were performed in the included patients. ROC curve analysis showed that a PCT value > 0.3 ng/mL was associated with the achievement of toxic voriconazole C_min_ > 3 mg/L, with sensitivity of 58.8% (95% CI 44.2–72.4%) and specificity of 71.8% (95% CI 62.4–80.0%). The AUC was 0.651 (95% CI 0.572–0.725; *p* = 0.0015; Figure 2).

In two cases, antifungal treatment was escalated to liposomal amphotericin B because of isolation of *Aspergillus niger* resistant to voriconazole in one case, and because of lack in attainment of effective voriconazole concentrations in the other case. The ICU mortality rate was 58.0%.

## 3. Discussion

To the best of our knowledge, this is the first study that described the impact of inflammatory status on voriconazole exposure in critically ill patients affected by CAPA. Interestingly, we found that both CRP and PCT thresholds may be significantly associated with an increased risk of attaining toxic voriconazole levels. Indeed, previous real-world evidence found that voriconazole C_min_ higher than 3 mg/L and 4 mg/L was associated with increased risk of hepatotoxicity and neurotoxicity, respectively [21,22].

Several factors may be responsible for the large inter- and intra-individual variability in voriconazole C_min_ commonly retrieved during treatment with standard fixed dosing regimens [18]. Specifically, voriconazole pharmacokinetics may be strongly affected by genetic polymorphisms of CYP2C19 isoenzyme, by drug–drug interactions with CYP2C9, 2C19, or 3A4 modulators, and/or by the inflammatory status [14,18,23,24]. In regard to the latter, several real-world studies carried out, mainly in patients with hematological malignancies undergoing hematopoietic stem cell/bone marrow transplantation and/or in solid organ transplant recipients, showed in the last decade that thresholds of inflammatory biomarkers like CRP or PCT may be significantly associated with toxic voriconazole levels [14]. Dote et al. [25] found in a cohort of 63 mixed patients (mainly hematological) that CRP values > 4.7 mg/dL were associated with an increased risk of voriconazole C_min_ exceeding 4 mg/L. In a case-control study including 266 hematological patients, Gautier-Veyret et al. [26] found at multivariate analysis that patients with CRP levels > 9.6 mg/dL showed a 27-fold higher risk of having voriconazole C_min_ ≥ 4 mg/L. Cheng et al. [27] found among 73 elderly patients that at ROC curve analysis serum PCT levels ≥ 1.31 ng/mL were associated with voriconazole C_min_ > 5 mg/L.

To the best of our knowledge, in the ICU scenario the impact of inflammation on voriconazole exposure was investigated previously only among 33 critically ill patients with sepsis or respiratory failure. The findings showed that a serum CRP level > 10 mg/dL was significantly associated with voriconazole C_min_ > 5.5 mg/L [28].

Overall, the findings are consistent with the CRP and PCT thresholds identified at ROC analysis in our homogeneous cohort of CAPA critically ill patients and may support the contention that voriconazole exposure is significantly affected by the degree of inflammation even in the novel scenario of CAPA among ICU admitted patients. Notably, two previous real-world experiences in COVID-19 patients affected by CAPA have raised the potential issue of the impact of inflammatory status on voriconazole exposure [29,30]. Le Daré et al. [29] described a case of critically ill COVID-19 patient affected by acute respiratory distress syndrome due to CAPA superinfection in which unexpected variations in voriconazole exposure were reported. Specifically, a remarkable impact of inflammatory degree on voriconazole C_min_ was observed, considering that the metabolite ratios (expressed as voriconazole N-oxide/voriconazole ratio) ranged from <0.3 during the inflammatory period to >1 after resolution of inflammation [29]. These findings were consistent with a significant decrease in CRP levels. Furthermore, a case series including 13 patients affected by CAPA (of which 12 required ICU admission) found higher voriconazole levels (in terms of C_min_) compared to those retrieved in 13 non-COVID-19 patients treated with voriconazole, leading to significantly higher transaminase levels [30]. Notably, a mild and positive correlation was found between voriconazole C_min_ and CRP levels [30]. However, in none of these real-world experience concerning CAPA scenario was a CRP or PCT threshold value predictive of a significant higher risk of voriconazole overexposure identified.

On this basis, implementing a TDM-based expert clinical pharmacological advice program for optimizing voriconazole exposure should become mandatory even in CAPA critically ill patients [20,31], as previously recommended for other settings [10,32]. In this scenario, the careful assessment of the degree of inflammation should be taken into account in the personalization of voriconazole therapy. The identification of specific thresholds based on serum levels of CRP, PCT, and IL-6 predictive for voriconazole overexposure could be associated with information on genetic polymorphisms of CYP2C19 isoenzyme and of clinically relevant drug–drug interactions in order to provide the best accurate voriconazole dosing adjustment when TDM results are interpreted.

Limitations of our study have to be recognized. The retrospective monocentric study design and the limited sample size should be acknowledged. The role of CYP2C19 genetic polymorphisms and of drug–drug interactions in affecting voriconazole exposure could not be ruled out. However, the absence of co-treatment with strong CYP450 inhibitors or inducers may suggest a minor role of drug–drug interactions. Finally, we recognize that assessing a more specific pro-inflammatory biomarker like IL-6 rather than CRP or PCT would have been more appropriate for addressing this issue. Unfortunately, available data on serum IL-6 values were limited in our study patients and this precluded us from performing ROC curve analysis, which was eventually useful at identifying a specific IL-6 threshold. Conversely, the fact that ours is the first real-life experience exploring the impact of inflammation on voriconazole exposure in a cohort of critically ill patients affected by CAPA is a point of strength.

## 4. Materials and Methods

All the critically ill COVID-19 patients who were treated with voriconazole intravenously because of probable or proven CAPA at the COVID ICU of the IRCCS Azienda Ospedaliero–Universitaria of Bologna between 1 January 2021 and 31 December 2022 were retrospectively assessed. Patients were included if probable or proven CAPA was identified according to the 2020 European Confederation of Medical Mycology (ECMM)/International Society for Human and Animal Mycology (ISHAM) consensus criteria [33], and if underwent at least one therapeutic drug monitoring (TDM) of voriconazole trough level (C_min_) with concomitant assessment of the inflammatory biomarkers CRP and PCT.

Demographic (age, sex, height, weight, body mass index (BMI), underlying diseases (i.e., obesity, cardiovascular disease, diabetes mellitus, chronic kidney disease, hepatic cirrhosis, cancer, immunosuppression)) and clinical/laboratory data (need for mechanical ventilation and vasopressors, implementation of ECMO, COVID-19 antiviral therapies, serum CRP and PCT) were retrieved for each patient. Obesity was defined as a body mass index (BMI) ≥ 30 kg/m^2^. Voriconazole dosage, concomitant medications, number of TDM assessment per patient, baseline GM index on BAL specimen, positive BAL cultures for *Aspergillus* spp., bacterial superinfections, and ICU mortality rate were also retrieved. Clinically relevant drug interactions of voriconazole with concomitant medication were defined according to the EMA guidelines [34]. Co-administered drugs potentially increasing voriconazole exposure were defined as strong inhibitors whether causing a > 5-fold increase in voriconazole plasma AUC values or ≥an 80% decrease in clearance; moderate inhibitors whether causing a >2-fold increase in voriconazole plasma AUC values or a 50–80% decrease in clearance; mild inhibitors whether causing a 1.25- to 2-fold increase in voriconazole plasma AUC values or a <50% reduction in clearance. Conversely, co-administered drugs potentially reducing voriconazole exposure were defined as strong, moderate, or mild inducers whether causing a >80%, a 50–80%, or a <50% decrease in voriconazole plasma AUC values.

Voriconazole was prescribed at the discretion of the treating physician or infectious disease consultant according to current clinical practice implemented at the IRCCS Azienda Ospedaliero-Universitaria of Bologna. Voriconazole was started at the recommended dosage of 6 mg/kg every 12 h for the first 24 h (loading dose), followed by 4 mg/kg every 12 h as a maintenance dose (MD), and was subsequently optimized by means of a real-time TDM-based clinical pharmacological advice strategy.

Blood samples for assessing plasma voriconazole C_min_ were collected 5–15 min before one drug administration after at least 48 h from starting treatment. Voriconazole blood concentrations were measured by means of a liquid chromatography-tandem mass spectrometry (LC–MS/MS) commercially available method (Chromsystems Instruments and Chemicals GmbH, Munich, Germany) [35]. TDM results were made available via the intranet to the MD clinical pharmacologist who provided an expert clinical pharmacological advice for prompt dosing adaptation by ICU physicians within the same day of TDM sampling. The desired voriconazole C_min_ range was set at 1.0–3.0 mg/L in agreement with international guidelines [10], and recent findings [21,22]. Two recent meta-analyses showed that voriconazole C_min_ > 3.0–4.0 mg/L may be associated with higher risk of hepatotoxicity [21,22].

Continuous data were presented as median and interquartile range (IQR), whereas categorial variables were expressed as count and percentage. The C/D was used as a surrogate marker of total clearance by dividing voriconazole C_min_ by daily dose per kg of body weight. ROC curve analysis was carried out by using on the one hand CRP or PCT values as the test variable, and on the other hand voriconazole C/D ratio > 0.375 (equivalent to a C_min_ value of 3 mg/L normalized to the MD of 8 mg/kg/day) as the state variable and defined as toxic level [22,36]. AUC along with 95% CI were calculated. The optimal cut-off point was computed by means of the Youden Index method. The Youden Index was calculated according to the following equation: sensitivity (%) + specificity (%) − 100. A *p* value < 0.05 was considered significant. Statistical analysis was performed by using the MedCalc for Windows (MedCalc statistical software, version 19.6.1, MedCalc Software Ltd., Ostend, Belgium). The study was approved by the Ethics Committee of IRCCS Azienda Ospedaliero-Universitaria of Bologna (n. 442/2021/Oss/AOUBo approved on 28 June 2021).

## 5. Conclusions

Our findings suggest that in critically ill patients with CAPA, CRP and PCT values above the identified thresholds may cause downregulation of voriconazole metabolism and favor voriconazole overexposure. This may lead to potentially toxic concentrations, which might cause hepato- and/or neuro-toxicity. In this scenario, a real-time TDM-based clinical pharmacological advice strategy should be implemented for granting optimal management of voriconazole treatment of CAPA. In clinical settings in which a real-time TDM strategy is unavailable, the identified CRP and PCT thresholds could be useful for providing clinicians a guide for promptly adopting voriconazole dosing adjustments in order to minimize the attainment of toxic concentrations. Large prospective clinical studies are warranted for confirming our findings and for assessing the potential impact on voriconazole efficacy and/or safety.

## Figures and Tables

**Figure 1 antibiotics-12-00764-f001:**
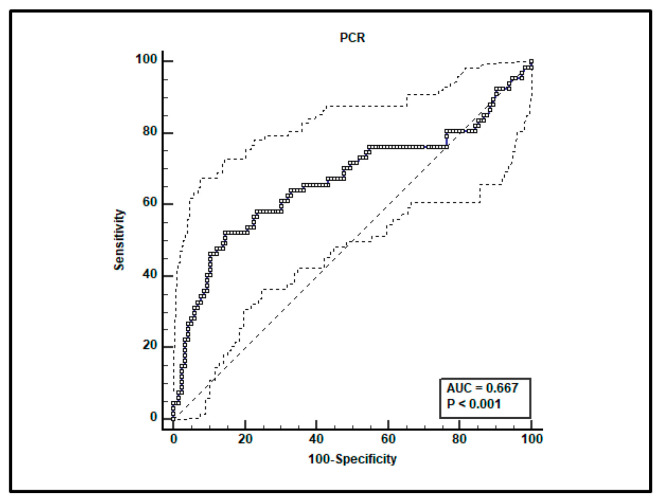
ROC curve analysis for voriconazole C_min_ > 3 mg/L. An optimal cut-off of CRP value > 11.46 mg/dL was found with a sensitivity of 52.2% and specificity of 85.2%.

**Figure 2 antibiotics-12-00764-f002:**
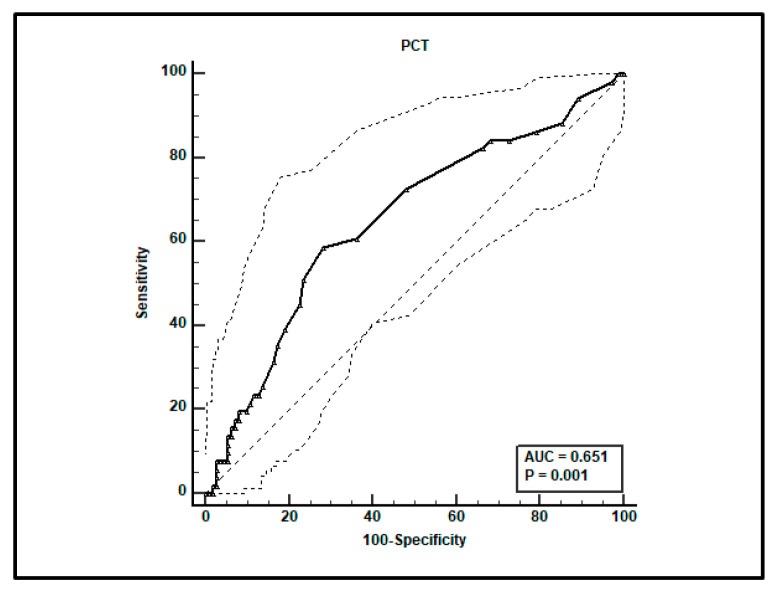
ROC curve analysis for voriconazole C_min_ > 3 mg/L. An optimal cut-off of PCT value > 0.3 ng/mL was found with a sensitivity of 58.8% and specificity of 71.8%.

**Table 1 antibiotics-12-00764-t001:** Demographic and clinical variables of critically ill patients undergoing at least one TDM assessment of voriconazole during treatment for COVID-associated pulmonary aspergillosis.

Demographics and Clinical Variables	Patients (*n* = 50)
Patient demographics	
Age (years)	65.5 (60.0–73.75)
Gender (Male/Female)	27/23 (54.0/46.0)
Body weight (kg)	77 (70–90)
Body mass index (kg/m^2^)	27.3 (24.3–30.9)
Comorbidities	
Obesity	15 (30.0)
Cardiovascular disease	7 (14.0)
Type 2 diabetes mellitus	10 (20.0)
Chronic kidney disease	2 (4.0)
Hepatic cirrhosis	0 (0.0)
Cancer	8 (16.0)
Transplantation	6 (12.0)
Severity of illness	
Mechanical ventilation ^1^	45 (90.0)
Septic shock ^1^	22 (44.0)
Extracorporeal membrane oxygenation ^1^	3 (6.0)
COVID-19 treatment	
Dexamethasone	44 (88.0)
Low molecular weight heparin	26 (52.0)
Tocilizumab	23 (46.0)
Remdesivir	11 (22.0)
Monoclonal antibodies	10 (20.0)
Colchicine	8 (16.0)
Molnupiravir	1 (2.0)
Nirmatrelvir/r	1 (2.0)
Microbiological data	
Baseline galactomannans on BAL	3.41 (1.84–4.63)
Patients with *Aspergillus* spp. growth at BAL cultures	9 (18.0)
*Aspergillus fumigatus*	5
*Aspergillus flavus*	2
*Aspergillus niger*	1
*Aspergillus fumigatus + flavus*	1
Bacterial superinfections	42 (84.0)
Gram-positive	
*Enterococcus faecalis*	8 (16.0)
*Enterococcus faecium*	5 (10.0)
*Methicillin-resistant Staphylococcus aureus (MRSA)*	4 (8.0)
*Methicillin-susceptible Staphylococcus aureus (MRSA)*	3 (6.0)
*Methicillin-susceptible Staphylococcus epidermidis (MSSE)*	1 (2.0)
*Methicillin-resistant Staphylococcus epidermidis (MRSE)*	1 (2.0)
Gram-negative	
*Pseudomonas aeruginosa*	12 (24.0)
*Klebsiella pneumoniae*	12 (24.0)
*Acinetobacter baumannii*	7 (14.0)
*Klebsiella aerogenes*	4 (8.0)
*Stenotrophomonas maltophilia*	3 (6.0)
*Escherichia coli*	2 (4.0)
*Enterobacter cloacae*	2 (4.0)
*Morganella morgannii*	1 (2.0)
*Proteus mirabilis*	1 (2.0)
*Enterobacter kobei*	1 (2.0)
*Klebsiella variicola*	1 (2.0)
Carbapenem-resistant isolates	11 (22.0)
Concomitant agents	
Patients concomitantly treated with inhibitors or inducers of CYP2C9, 2C19 and/or 3A4	34 (68.0)
Dexamethasone	12 (24.0)
Omeprazole	11 (22.0)
Dexamethasone + omeprazole	10 (20.0)
Phenytoin + omeprazole	1 (2.0)
Voriconazole treatment	
Average C_min_ (mg/L)	2.47 (1.75–3.33)
Average daily dose (mg/kg)	7.69 (6.99–8.13)
Average concentration/dose ratio	0.29 (0.14–0.46)
Inflammatory biomarkers	
Average CRP levels during voriconazole therapy (mg/dL)	7.36 (3.52–16.61)
Average PCT levels during voriconazole therapy (ng/mL)	0.38 (0.10–1.03)
Clinical outcome	
Need to antifungal treatment escalation to L-AmB	2 (4.0)
ICU mortality	29 (58.0)

Data are presented as median (IQR) for continuous variables and as *n* (%) for dichotomous variables. BAL: bronchoalveolar lavage; C_min_: trough concentrations. CRP: C-reactive protein; ICU: intensive care unit; L-AmB: liposomal amphotericin B; PCT: procalcitonin. ^1^ At the start of voriconazole treatment.

## Data Availability

The data presented in this study are available on request from the corresponding author. The data are not publicly available due to privacy concerns.

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
