# Peer review of "Impact of Inflammation on Voriconazole Exposure in Critically ill Patients Affected by Probable COVID-19-Associated Pulmonary Aspergillosis"

_antibiotics, 2023, doi:10.3390/antibiotics12040764_

Round 1

Reviewer 1 Report

Dear authors,

The manuscript is a well written one.

Here some suggestions to improve it.

1. The authors should insert the statistical data of to the inflamatory markers CRP or PCT value associated with toxic voriconazol levels. It will be much easier to follow.

2. The manuscript lacks the Conclusion part. Please write the conclusion keeping in the mind to elaborate your findings in terms of their utility for society, researchers.

Thank you!

Regards!

Author Response

Dear authors,

The manuscript is a well written one.

Here some suggestions to improve it.

We thank the reviewer for appreciating our manuscript.

Q1. The authors should insert the statistical data of to the inflamatory markers CRP or PCT value associated with toxic voriconazol levels. It will be much easier to follow.

A1. We agree with this suggestion, but respectfully highlight that the statistical data concerning which are the best CRP and PCT thresholds associated with voriconazole toxicity were already mentioned in the Results section (refer to Line 110-115 and 118-123) and in Figure 1 and 2 legends.

Q2. The manuscript lacks the Conclusion part. Please write the conclusion keeping in the mind to elaborate your findings in terms of their utility for society, researchers.

Thank you!

Regards!

A2. We thank the reviewer for this suggestion. We moved the last paragraph of the discussion to the novel Conclusions section, discussing the potential application of our findings in daily clinical practice (refer to Line 261-271).

Reviewer 2 Report

Dear Authors,

This study is original and provides us with valuable data on the toxicity of voriconazole in critically ill patients. The TDM method is adequately applied and may be necessary for the administration of voriconazole in these patients in future. I have suggestions to expand the Results section:

1. To perform ROC curve analysis by using the role of elevated concentrations of IL-6.  

2. To compare the efficacy and toxicity of voriconazole in patients who received dexamethasone or dexamethasone+tocilizumab or omperazole.

3. To show whether these patients had comorbidities (obesity, T2DM, cancer, renal insufficiency or hepatic insufficiency) and see the effectiveness and toxicity of voriconazole in those groups of patients, if it is possible.

4. To show whether the patients had infections with multiresistant bacteria and to see the outcome of the disease in those groups of patients.

 With respect,

Author Response

Reviewer #2

Dear Authors,

This study is original and provides us with valuable data on the toxicity of voriconazole in critically ill patients. The TDM method is adequately applied and may be necessary for the administration of voriconazole in these patients in future. I have suggestions to expand the Results section:

We thank the reviewer for appreciating our study.

Q1. To perform ROC curve analysis by using the role of elevated concentrations of IL-6. 

A1. We thank the reviewer for this useful suggestion. Although performing ROC curve analysis by assessing the role of IL-6 represented one of our primary aims, unfortunately the available data on IL-6 values were limited for our study patients and this prevented us from performing it. We specified this issue in the Limits section (refer to Line 192-194).

Q2. To compare the efficacy and toxicity of voriconazole in patients who received dexamethasone or dexamethasone+tocilizumab or omperazole.

A2. We thank the reviewer for this relevant comment and suggestion. Although we recognize the importance of assessing voriconazole efficacy and toxicity according to different concomitant medications, we respectfully highlight that this aim is out of scope of our study. The primary aim of our study was to explore the impact of the degree of inflammation on voriconazole exposure in critically ill patients affected by CAPA. In this context, the assessment of the relationship with voriconazole efficacy and toxicity falls outside the scope of our study. However, we added a dedicated sentence in the Conclusions section highlighting the importance of assessing the impact of our findings on voriconazole efficacy and/or safety (refer to Line 270-271).

Q3. To show whether these patients had comorbidities (obesity, T2DM, cancer, renal insufficiency or hepatic insufficiency) and see the effectiveness and toxicity of voriconazole in those groups of patients, if it is possible.

A3. We thank the reviewer for this comment. We added comorbidities in Results section (refer to Line 89-90 and Table 1). However, as previously mentioned in response to comment no. 2, the assessment of the relationship with voriconazole efficacy and toxicity is out of scope of our study.

Q4. To show whether the patients had infections with multiresistant bacteria and to see the outcome of the disease in those groups of patients.

 With respect.

A4. We thank the reviewer for this comment. We added bacterial superinfections in Results section (refer to Line 100-103 and Table 1). However, as previously mentioned in response to comment no. 2, the assessment of the relationship with voriconazole efficacy and toxicity is out of scope of our study.

Round 2

Reviewer 2 Report

Authors have corrected the work adequately. They included in the text the limits of the study. I believe that the paper can be accepted for publication.
ri have repaired the work adequately. They included in the text the limits of the study. I believe that the paper can be accepted for publication.